# Modification of SPWM Modulating Signals for Energy Balancing Purposes

**Yesenia Reyes-Severiano** [1] , **Susana Estefany De León Aldaco** [1,*] , **Jesus Aguayo Alquicira** [1] , **Luis Mauricio Carrillo-Santos** [1] , **Ricardo Eliú Lozoya-Ponce** [2] and **Jesús Alfonso Medrano Hermosillo** [2]

1   Electronic Engineering Department, TecNM/Cenidet, Cuernavaca 62490, Mor, Mexico
2   Instituto Tecnológico de Chihuahua (ITCH), Chihuahua 31310, Chih, Mexico
*   Correspondence: susana.da@cenidet.tecnm.mx

**Abstract:** There is currently a growing interest in efficient power generation and transformation without increasing the cost and complexity of a system. One type of transformation is the conversion of energy from a direct current to an alternating current, which is used in various applications, for example, in photovoltaic systems. One of the elementary components of these systems is the inverter. However, there are several drawbacks in the design of these systems, such as the energy balance between their semiconductor devices. Therefore, it is important to study alternatives to balance the energy and thus achieve a positive impact on both the economic and reliability aspects of the system. This article deals with the study, design, and implementation of the modification of a modulation technique with reconstructed modulating signals, which aims to ensure the energy balance in each cell of a multilevel inverter and, at the same time, present better results concerning the different parameters of comparison established, such as harmonic distortion, percentage of unbalance, percentage of use of digital resources, and power transferred to the load.

**Keywords:** cascaded; multilevel inverter; modulation technique; modulation index; total harmonic distortion; energy balance; energy

## 1. Introduction

Currently, there are several applications in which the transformation of a direct current (DC) to an alternating current (AC) conversion is used. One of them is in photovoltaic installations for electric power generation, which is a growing trend, both globally and nationally [1,2]. The main reasons why renewable energies have established themselves as important sources of electrical energy include the increased cost-effectiveness of renewable technologies, policy initiatives, improved access to financing, energy security, environmental support, the need for access to modernized energy, and the need for more efficient uses of resources [1–4].

Figure 1 shows a general schematic block diagram that includes the main elements that make up an isolated photovoltaic system. It is worth mentioning that there are other elements related to this system, such as an MPPT tracker and an output filter or link capacitor.

Of the above elements, this research work focuses on the DC–AC converter, called the inverter. The inverters used in photovoltaic systems must mainly comply with characteristics such as reliability, power-boosting capacity, and cost-effectiveness, and they must inject a sinusoidal current into the power grid (in the case of interconnected systems) [5,6].

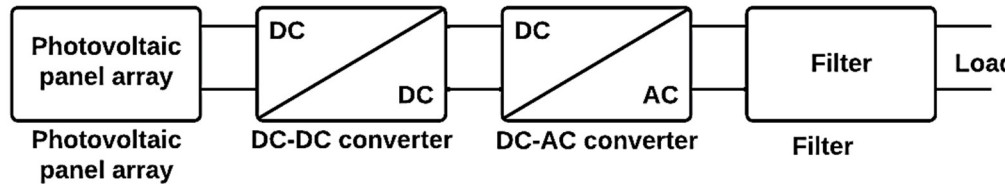

**Figure 1.** Diagram of the main elements of a photovoltaic system.

Multilevel inverters have advantages over conventional inverters, such as reduced switching losses, reduced stress on semi-conductor power devices, reduced total harmonic distortion (THD), and lower power consumption [7,8]. Among the multilevel topologies, the cascaded H-bridge multilevel inverter topology (CHBMI) stands out, which has the advantage of having individual sources, which in this case are strings of photovoltaic panels, thus reducing the nominal voltage of the topology's power semiconductor devices.

According to [9], the parameters for comparison between the inverters used in photovoltaic panels are the number of components used, component stress, and nominal power of the system. In addition, to evaluate the performance of PV inverters, the following parameters can be highlighted:

- efficiency;
- power density;
- installation cost;
- leakage current minimization;
- maximum power transfer (MPPT); and
- energy balance (voltage, current, and power).

In the design of inverters, some problems affect the reliability of the system [4,10]. For example, a CMLI is designed in a modular way, and it is enough to design a single cell or H-bridge belonging to the topology and then reproduce it for the other existing cells (depending on the number of levels of the inverter). This type of modular design is performed considering the maximum power transfer, and doing it this way ensures all the cells are designed under the same voltage and current parameters, assuming all the complete bridges will handle the same power, which is not entirely true since most of the time, this depends on the modulation technique used.

If the design is not done in a modular way, each cell would have to be designed independently, considering the power transferred to the load by each cell. Therefore, it is required that the design of power semiconductor devices be carried out for different current ratings. Figure 2 shows a cell that transfers the most power with a larger box, while the smaller box represents a cell with the least power transfer.

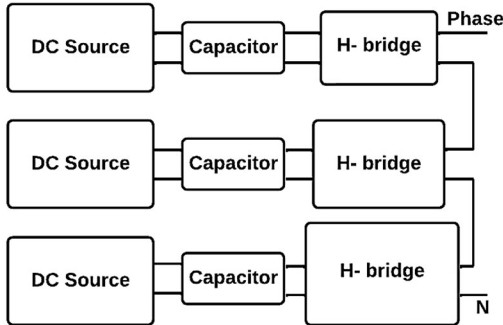

**Figure 2.** Independent cell construction considering the power transferred to the load.

Designing a converter in this independent way provides an increase in the cost of the system, both economically and intellectually, which brings us back to the modular construction of the converter. However, this type of design also has disadvantages because, if each cell transfers a different power to the load, it causes the following:

- decreased time between failures in the cells that handle higher power levels;
- higher stress (current, voltage, and temperature) produced in semi-conductor power devices in the cells that handle higher power levels;
- higher stress (current, voltage, and temperature) produced in semi-conductor power devices in cells handling higher power levels; and
- oversizing in the cells that handle lower power levels since the design is made considering the maximum power.

The non-uniform distribution of energy causes the above problems to directly affect the economics and reliability of the system.

This paper addresses an alternative to preserve the modular design and to not reach the independent design of each cell by modifying the embedded modulation technique to switch to the multilevel inverter. The objective is to achieve an energy balance in each cell and to present improvements in the comparison parameters established with respect to the existing strategies.

The rest of the article is presented as follows: Section 2 defines the stages that make up the system under study, and the design of the alternative modulation strategy "reconstructed modulators" is presented. Subsequently, Section 3 presents the results obtained experimentally using the modulation variables together with the power stage. The analysis and discussion related to the experimentally obtained results are addressed in Section 4. Finally, Section 5 contains the conclusions reached in the development of this work.

## 2. Materials and Methods

This section describes the stages into which the study system was divided: power and modulation. The power stage deals with the cascade multilevel topology used and the modulation stage deals with conventional modulation techniques used, as well as those that carried out the energy balancing to make a comparison with the alternative strategy proposed in this article, whose design is described in the end of this section.

### 2.1. Power Stage

The topology used for the power stage is the seven-level, single-phase cascaded multilevel inverter, which has three single-phase full bridges connected in a cascade (Figure 3). This topology has the following advantages for its application:

- capable of handling high output voltage and power levels;
- independent power supplies;
- a low harmonic content in the output voltage; and
- the power semiconductor devices support only the voltage present in a DC source.

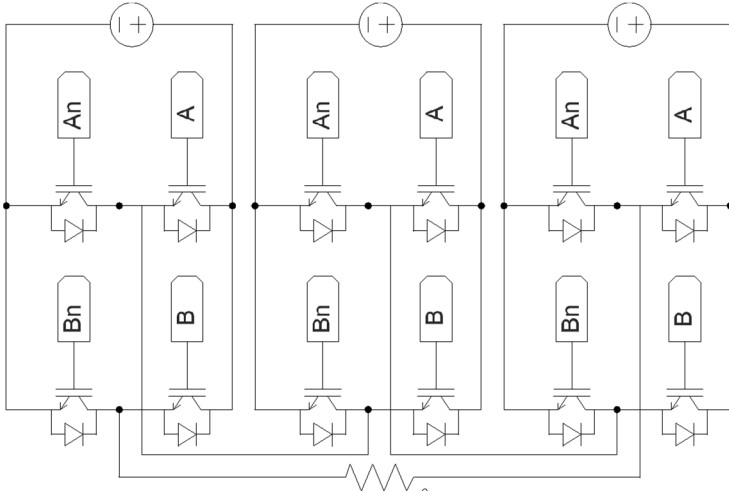

**Figure 3.** Schematic diagram of the power stage: CHBMI and resistive load.

Table 1 shows the design specifications.

**Table 1.** Topology design specifications used.

| Parameter | Value |
|---|---|
| DC source | 120 V |
| Modulating signal frequency | 3000 Hz |
| Carrier signal frequency | 60 Hz |
| Number of output voltage levels | 7 |
| Number of cells per phase | 3 |
| Modulation index | 0.9 |
| Resistive load | 260 Ω |

*2.2. Modulation Stage*

To address the modulation stage, the carrier-arrangement PWM strategy is highlighted [6,11–13]. Other strategies perform modifications of this strategy for different target functions, which is the case with existing modulation techniques that perform energy balancing. Therefore, the modulation techniques used in this study are divided into three groups, the third being the strategy proposed in this article. It is worth mentioning that the different groups are discussed to appreciate the comparison between the different strategies.

2.2.1. Strategy of Existing Modulation without Energy Balance Purposes

The non-energy-balancing modulation strategy used in this article is Phase Disposition (PD), which is governed by the principle of multicarrier strategies, consisting of the comparison of a given number of carrier signals which have the same amplitude, concerning a reference modulating signal. The number of carrier signals ($S_c$) necessary to generate a certain number of voltage levels at the output (n) can be determined by employing (1):

$$S_c = n - 1,$$ (1)

where $S_c$ is the number of carrier signals required and n is the number of voltage levels at the output.

It is important to emphasize that the PD strategy is a variant of the multicarrier PWM technique. Most variants of this technique are not used for energy balancing purposes, such as: Phase Opposite Disposition (POD), Alternative Phase Opposite Disposition (APOD), which follows the same principle described above but varies by the phase of the carrier signals, and each of them has a DC increment [14]. These variants are presented below:

- PD: in this modulation strategy, the carrier signals are in phase with each other.
- POD: in this modulation strategy, the carrier signals are 180° out of phase with respect to the adjacent carrier signal.
- APOD: in this modulation strategy, the carrier signals above zero are 180° out of phase with respect to the carrier signals below zero.

Figure 4 shows the three variants of the carrier disposition PWM modulation strategy. Each modulation technique presented in this figure has six carrier signals when compared with the modulating signals that generate an output wave of seven levels. The carrier signals have the same amplitudes but different displacements.

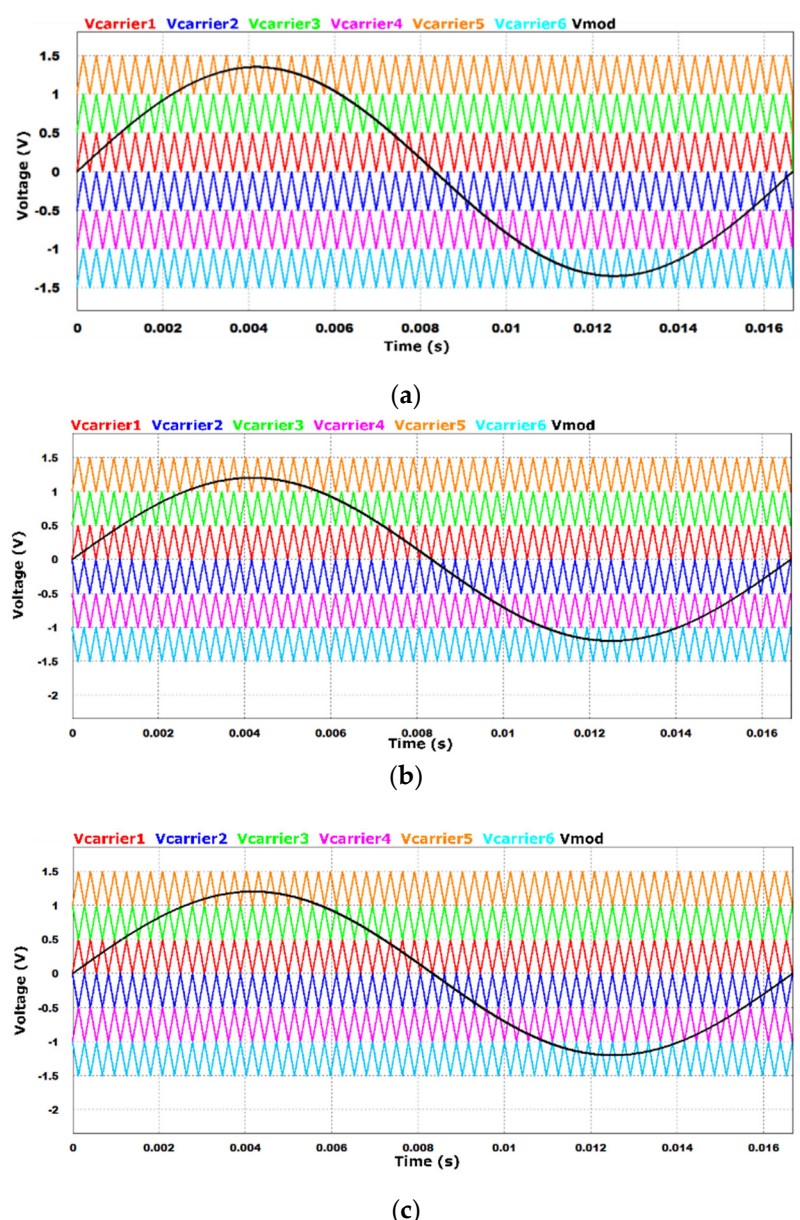

**Figure 4.** Variants of the PWM modulation strategy of carrier disposition: (**a**) PD, (**b**) POD, and (**c**) APOD.

2.2.2. Existing Modulation Strategies for Energy Balancing Purposes

The previous section dealt with modulation strategies without energy balance purposes. We proceeded to address the techniques whose objective function is to carry out the energy balance between the cells that make up a cascaded multilevel inverter. This identified areas of opportunity for improvement under which the proposed alternative strategy is governed. We used the above to finally carry out the comparison between these existing strategies and the alternative strategy proposed in this paper.

Three main strategies to perform energy balancing stand out [6,11,15–17], which are: PSC PWM (phase shift carrier PWM) and LS PWM (level shifted PWM), with a carrier level shift, in two variants: a per modulator signal cycle (LS modulator) and a per carrier signal cycle (LS carrier) [13,18–21].

These strategies are modified variants of the multi-carrier PWM modulation technique, which has the main purpose of creating output voltage harmonics at high frequencies around the switching frequency. Each variant modifies the switching sequence, aiming at the energy balance between the cells of the multilevel topology, and they are also directly related to the quality of the output signal, as presented below.

- PSC PWM [17,22–24]

In this modulation strategy, the carrier signals are of the same amplitude and frequency as each other. However, they have phase shifts between them, and this to position, the switching ripple is of a higher frequency than the switching frequency, depending on the number of carrier signals used. To determine the angle of phase shift between the carriers, (2) is used:

$$\varphi = \frac{360°}{2n} \tag{2}$$

where $\varphi$ is the phase shift angle of the carrier signal and $n$ is the number of signals per phase.

Figure 5 shows a close-up of the carrier signals of this technique, which presents phase shifts between them of 60°. The frequency of the carrier signals present in Figure 5 was reduced six times, which allows for a better appreciation of the phase difference between them.

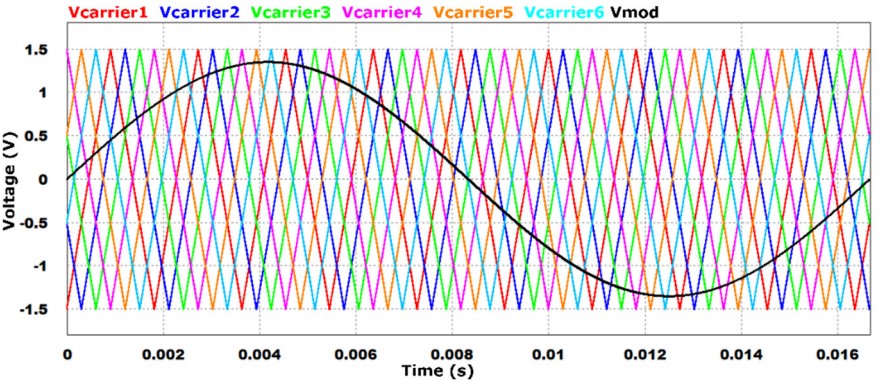

**Figure 5.** PSC PWM modulation strategy carrier signals.

- LS PWM per modulating signal cycle [22,25–28]

In this modulation strategy, the carrier signals are shifted in level according to the modulating signal cycles to perform the energy balance [29]. Figure 6 shows the carrier and modulator signals of this technique. The carrier signals are shifted in level according to the modulating signal cycles to perform the energy balance.

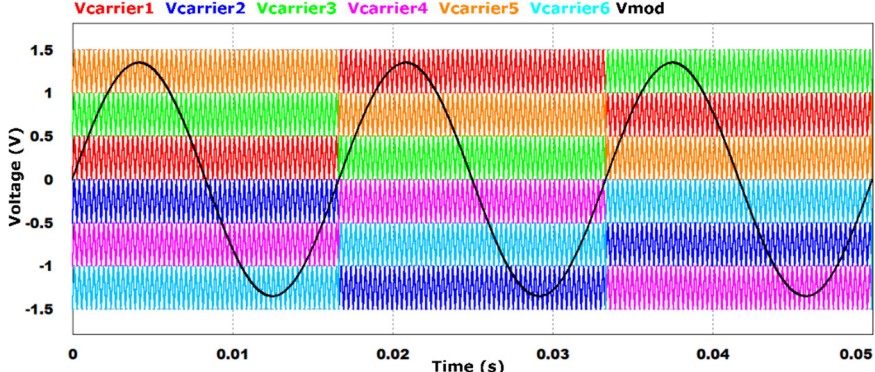

**Figure 6.** Carrier and modulator signals of the LS PWM modulation strategy, per the modulating signal cycle.

The number of cycles required to perform the energy balance is half the number of carrier signals; that is, if the inverter is seven levels, then the energy balance will be performed after three modulating signal cycles.

- LS PWM per carrier signal cycle [6,23,28,29]

In this strategy, the carrier signals are shifted in level according to each cycle of the same. It is worth mentioning that this modulation technique, as with the previous one presented, uses the multicarrier PD PWM strategy as a basis and modifies it.

Figure 7 shows the carrier signals and a modulating signal of the LS PWM technique. It can be seen that the carrier signals are shifted in level according to each cycle of the same carrier, which leads to increased switching losses.

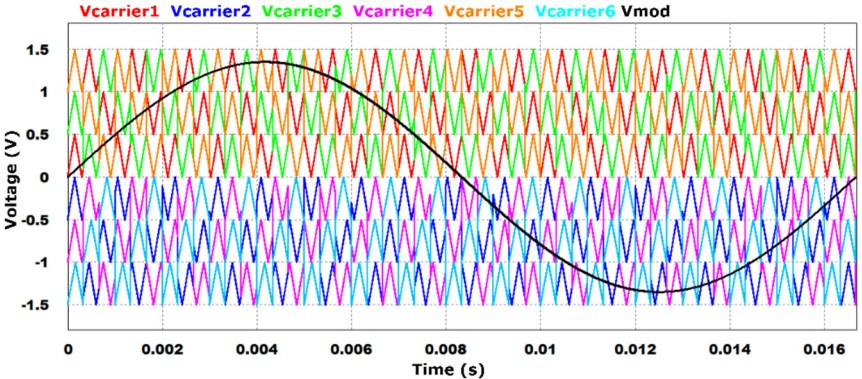

**Figure 7.** Carrier signals and a modulator signal of the LS PWM modulation strategy, per the carrier signal cycle.

### 2.2.3. Proposed Alternative Modulation Strategy, "Reconstructed Modulators"

Considering the three strategies for energy balance purposes discussed in the previous section, the comparison parameters were established between them to obtain key points that would lead to an improvement in the design of a proposed alternative strategy.

The comparison parameters established under which the alternative modulation strategy is governed include the following:

- energy balance between multilevel converter cells
- percentage of unbalance
- harmonic content
- computational expense in implementation
- reduction in the number of carrier signals per level

The proposed modulation strategy arose from the principle of ensuring that the voltage of each one is the same, and the cells must conduct the same time between them and have equal switching losses, all of which keeps the equal relationship in the use of the amplitude of the sine wave at the time of comparison.

As analyzed, the existing strategies that perform energy balance are based on a modification of the carrier signals, keeping the modulating signal intact. Therefore, an alternative strategy is shown, and its main focus is not to modify the carrier signals, but rather, to modify the modulating signal, and at the same time, allow us to reduce the number of carrier signals required for modulation. This strategy modifies a signal without losing sight of the relationship between the area of the modulating signal that belongs to each triangular signal and the conduction time [27,30], and at the same time, obtains a minimum THD.

This modulation strategy requires the use of only two carrier signals and three modulating signals to obtain a single-phase output voltage signal of seven levels. The signals used in this strategy are the following:

- Carrier signal 1: fixed-amplitude triangular signal located in the positive quadrant (see (3))
- Carrier signal 2: fixed-amplitude triangular signal located in the negative quadrant (see (4))

$$V_{c1} = A_T \left[ \frac{2}{\pi} \arcsin \left( \sin \left( 2\pi f_c t - \frac{\pi}{2} \right) \right) \right] + A_T \tag{3}$$

$$V_{c2} = A_T\left[\frac{2}{\pi}arcsin\left(sin\left(2\pi f_c t - \frac{\pi}{2}\right)\right)\right] - A_T \tag{4}$$

where $V_{C1}$ and $V_{C2}$ are the voltage signals of the triangular carriers, $t$ is the period, $f_C$ is the frequency of the carrier signal, and $A_T$ is the amplitude of the triangular waveform.

- Modulating signal 1: A Sine signal is constructed in sections starting at an angle of $0°$. The angles ($\alpha$) are added, which set the level limits corresponding to each reconstructed modulator (see (5)). Modulating signal 1 is the one that, when compared with the two carrier signals, obtains the switching signals of cell 1. The amplitude varies depending on the modulation index.

$$V_{mod1} = \begin{cases} V_{part1} = \begin{cases} V_{part2_1} = V_m \sin(2\pi f_m t + \alpha_1) + \frac{2V_m}{3} & 0\langle t\rangle\alpha_1 \\ V_{part2_2} = \frac{V_m}{3} & \alpha_1\langle t\rangle\alpha_2 \\ V_{part2_3} = V_m \sin(2\pi f_m t + \alpha_2) & \alpha_2\langle t\rangle\pi \end{cases} \\ V_{part2} = \begin{cases} V_{part3_1} = V_m \sin(2\pi f_m t) + \frac{2V_m}{3} & 0\langle t\rangle\alpha_1 \\ V_{part3_2} = \frac{V_m}{3} & \alpha_1\langle t\rangle\pi - \alpha_1 \\ V_{part3_3} = V_m \sin(2\pi f_m t) + \frac{2V_m}{3} & \pi - \alpha_1\langle t\rangle\pi \end{cases} \\ V_{part3} = V_m \sin(2\pi f_m t + \alpha_2) + \frac{2V_m}{3} \end{cases} \tag{5}$$

where $V_{mod1}$, $V_{mod2}$, and $V_{mod3}$ are the voltages of cells 1, 2, and 3, respectively, $f_m$ is the modulating frequency of the sine waveform, $V_m$ is the amplitude of the sine waveform, and $V_{part}$ is the voltage for each section of the modulator used.

- Modulating signal 2: A sine signal is constructed in sections, starting at an angle of $120°$. The angles ($\alpha$) are added, which set the level limits corresponding to each reconstructed modulator (see (6)). Modulating signal 2 is the one that, when compared with the two carrier signals, obtains the switching signals of cell 2. The amplitude varies depending on the modulation index.

$$V_{mod2} = \begin{cases} V_{part1} = \begin{cases} V_{part2_1} = V_m \sin(2\pi f_m t + \alpha_1 + 120°) + \frac{2V_m}{3} & 0\langle t\rangle\alpha_1 \\ V_{part2_2} = \frac{V_m}{3} & \alpha_1\langle t\rangle\alpha_2 \\ V_{part2_3} = V_m \sin(2\pi f_m t + \alpha_2 + 120°) & \alpha_2\langle t\rangle\pi \end{cases} \\ V_{part2} = \begin{cases} V_{part3_1} = V_m \sin(2\pi f_m t + 120°) + \frac{2V_m}{3} & 0\langle t\rangle\alpha_1 \\ V_{part3_2} = \frac{V_m}{3} & \alpha_1\langle t\rangle\pi - \alpha_1 \\ V_{part3_3} = V_m \sin(2\pi f_m t + 120°) + \frac{2V_m}{3} & \pi - \alpha_1\langle t\rangle\pi \end{cases} \\ V_{part3} = V_m \sin(2\pi f_m t + \alpha_2 + 120°) + \frac{2V_m}{3} \end{cases} \tag{6}$$

- Modulating signal 3: A sine signal is constructed in sections, starting at an angle of $240°$. The angles ($\alpha$) are added, which set the level limits corresponding to each reconstructed modulator (see (7)). The modulating signal 3 is the one that, when compared with the two carrier signals, obtains the switching signals of cell 2. The amplitude varies depending on the modulation index.

$$V_{mod2} = \begin{cases} V_{part1} = \begin{cases} V_{part2_1} = V_m \sin(2\pi f_m t + \alpha_1 + 240°) + \frac{2V_m}{3} & 0\langle t\rangle \alpha_1 \\ V_{part2_2} = \frac{V_m}{3} & \alpha_1 \langle t\rangle \alpha_2 \\ V_{part2_3} = V_m \sin(2\pi f_m t + \alpha_2 + 240°) & \alpha_2 \langle t\rangle \pi \end{cases} \\ V_{part2} = \begin{cases} V_{part3_1} = V_m \sin(2\pi f_m t + 240°) + \frac{2V_m}{3} & 0\langle t\rangle \alpha_1 \\ V_{part3_2} = \frac{V_m}{3} & \alpha_1 \langle t\rangle \pi - \alpha_1 \\ V_{part3_3} = V_m \sin(2\pi f_m t + 240°) + \frac{2V_m}{3} & \pi - \alpha_1 \langle t\rangle \pi \end{cases} \\ V_{part3} = V_m \sin(2\pi f_m t + \alpha_2 + 240°) + \frac{2V_m}{3} \end{cases} \quad (7)$$

To generate the modulating signals, a sinusoidal signal is modified by dividing it into six parts, according to the number of carriers (three per half-cycle). These divisions will work at different times (Figure 8). Once the signal is divided, a modulator is built using the three parts, but at different times, obtaining the modulating signal shown in Figure 9.

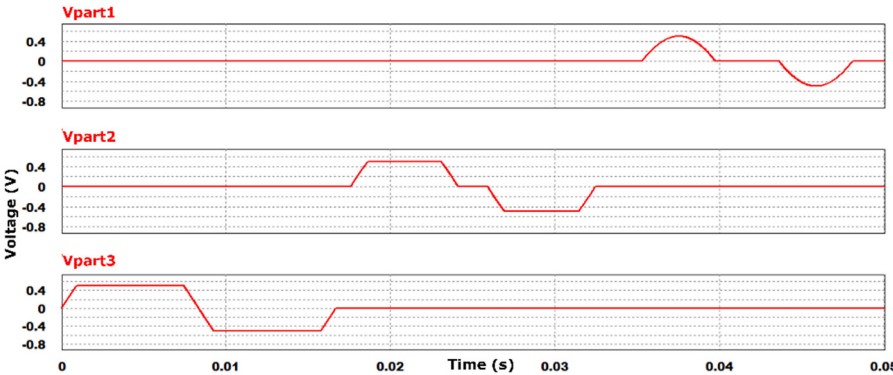

**Figure 8.** Chopped sinewave signal.

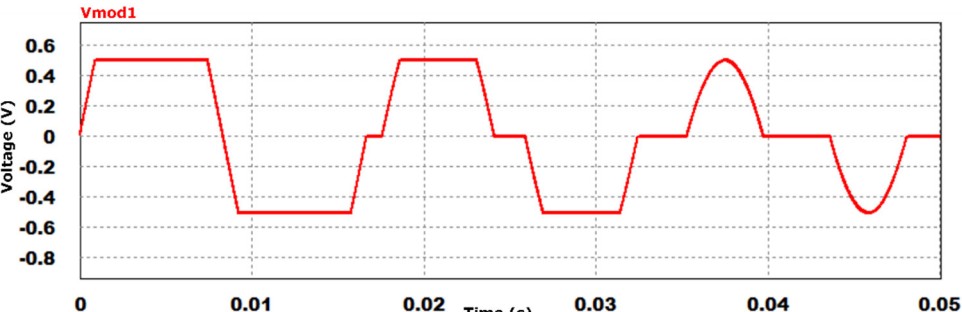

**Figure 9.** Modulating signal 1.

Two more modulating signals are constructed for this modulation, which are 120° out of phase, as shown in Figure 10. Using three low-frequency modulators, only two high-frequency carrier signals are required for the comparison (Figure 11).

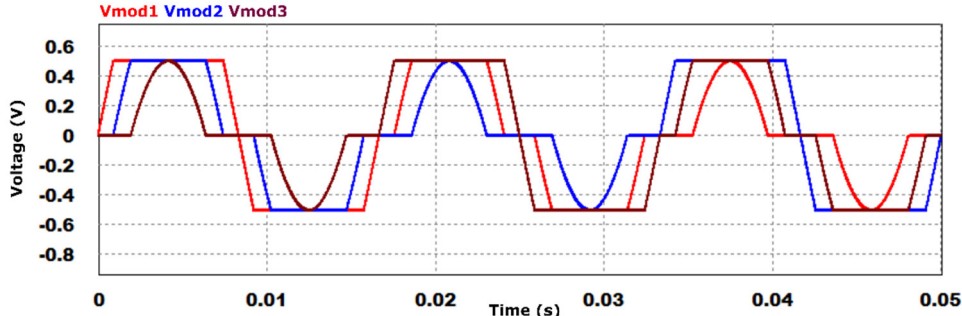

**Figure 10.** Modulating signals 1, 2, and 3.

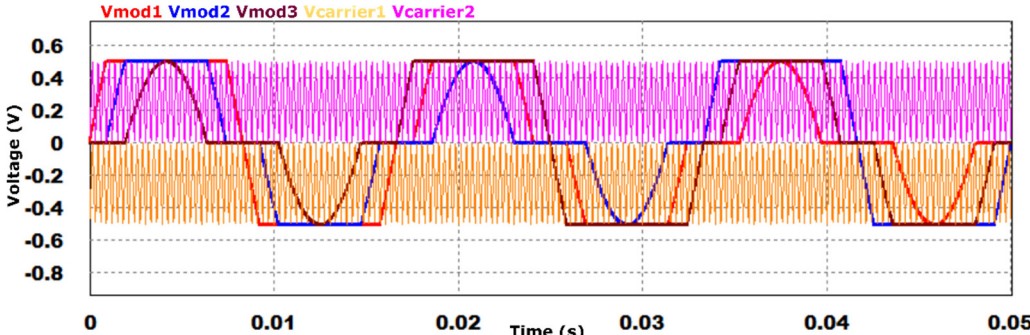

**Figure 11.** Reconstructed modulating signals and carrier signals.

Figure 12 shows the inverter output voltage, which shows that the seven required levels are obtained, obtaining a THD of 3.12% in the simulation.

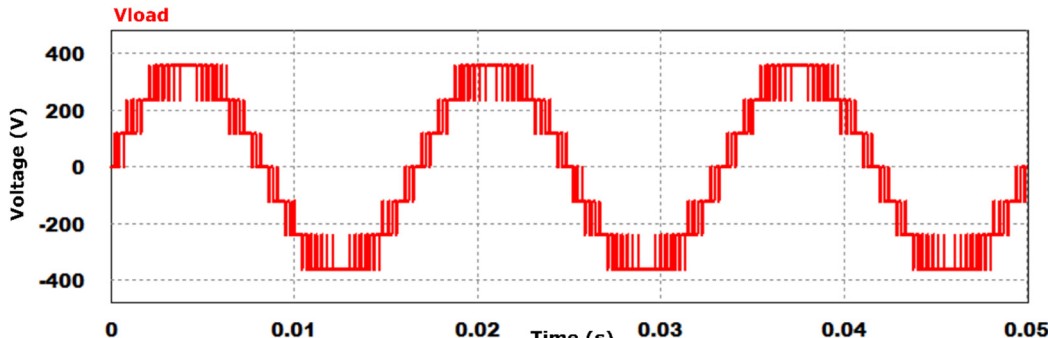

**Figure 12.** Output voltage.

## 3. Results

This section deals with the results obtained experimentally. Figure 13 shows the experimental platform developed. The power platform consists of two stages:

- Stage 1: in order to ensure that the DC supply voltage is the same as that of the H-bridges, the AC supply voltage of the network is first transformed, and then rectified to DC; and
- Stage 2: once the power balance of the three DC sources is ensured, the single-phase cascaded multilevel inverter is used to perform the DC to AC conversion, which is integrated by IRAMS10UP60b modules.

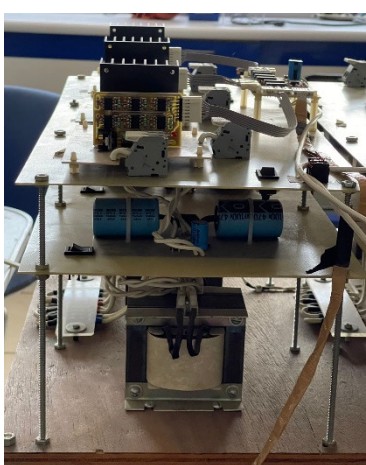

**Figure 13.** Experimental platform, multilevel topology.

Figure 14a,b shows a close-up of the implementation of stage 1 and stage 2, respectively.

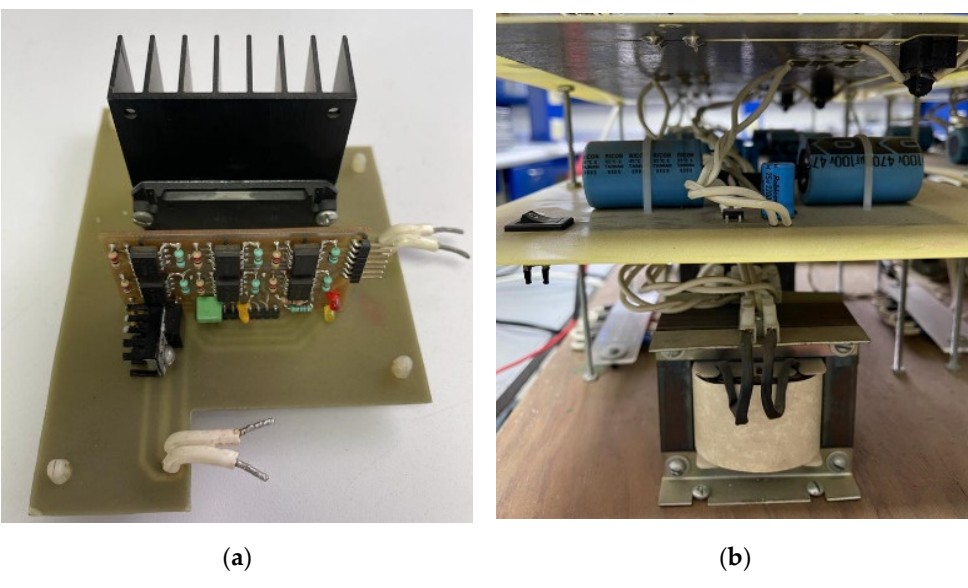

(**a**)                                               (**b**)

**Figure 14.** Power platform: (**a**) stage 1 and (**b**) stage 2.

For the modulation stage, the five strategies discussed in the previous section were implemented through the development of a nested VHDL code, implemented in Altera® Cyclone II FPGA. Figure 15 shows the implemented commutation signals of the different modulation strategies.

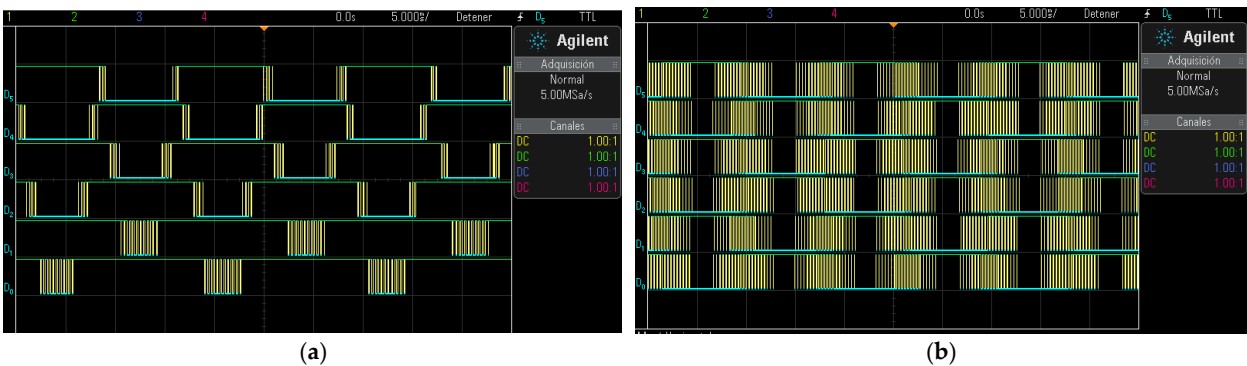

(**a**)                                               (**b**)

**Figure 15.** *Cont*.

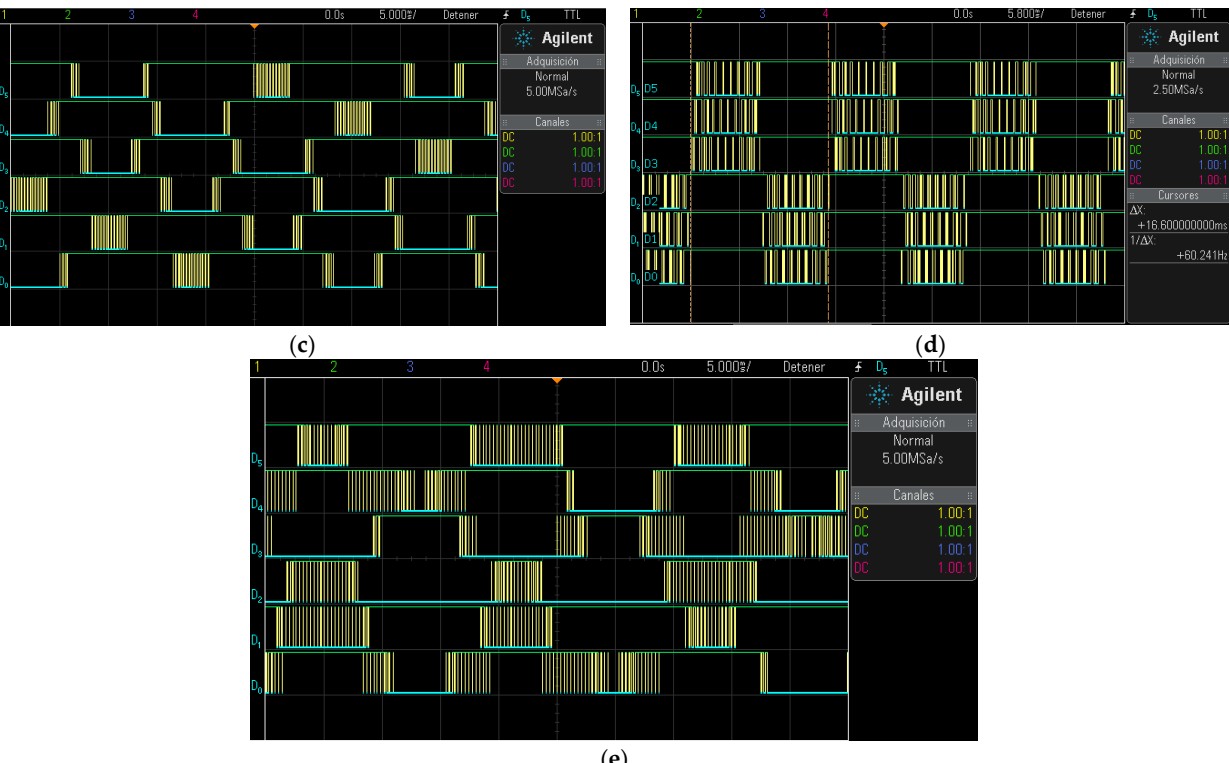

(**c**)

(**e**)

**Figure 15.** Implemented switching signals of the strategies: (**a**) PD PWM; (**b**) PSC PWM; (**c**) LS PWM per modulating signal cycle; (**d**) LS PWM per carrier signal cycle; and (**e**) proposed alternative modulation strategy, "reconstructed modulators".

The above digital signals are used to activate/deactivate the switches of the multilevel inverter of the power stage, thus obtaining a step voltage at the output. Figure 16 shows the voltage signal at the inverter output using the modulation strategy proposed in this article.

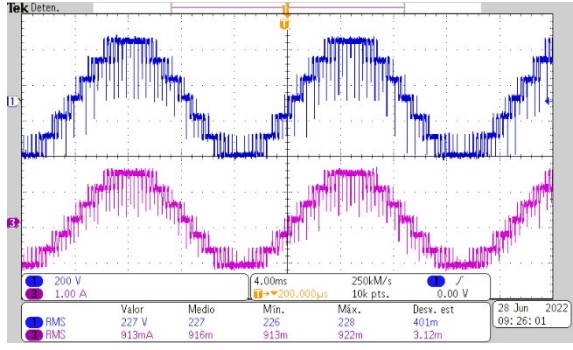

**Figure 16.** Signals of voltage (blue) and current (purple) in the cells at the output of the multilevel inverter using the alternative strategy, "reconstructed modulators".

In addition, the power and energy signals were obtained at the output of each cell belonging to the multilevel inverter. This was done to observe the unbalance or balance between them, depending on the strategy used, as well as to carry out the comparison of the different study parameters to finally verify the behavior of the alternative strategy proposed in this article.

Figure 17 shows the voltage, current, and power signals obtained in each cell using the proposed alternative modulation strategy, "reconstructed modulators".

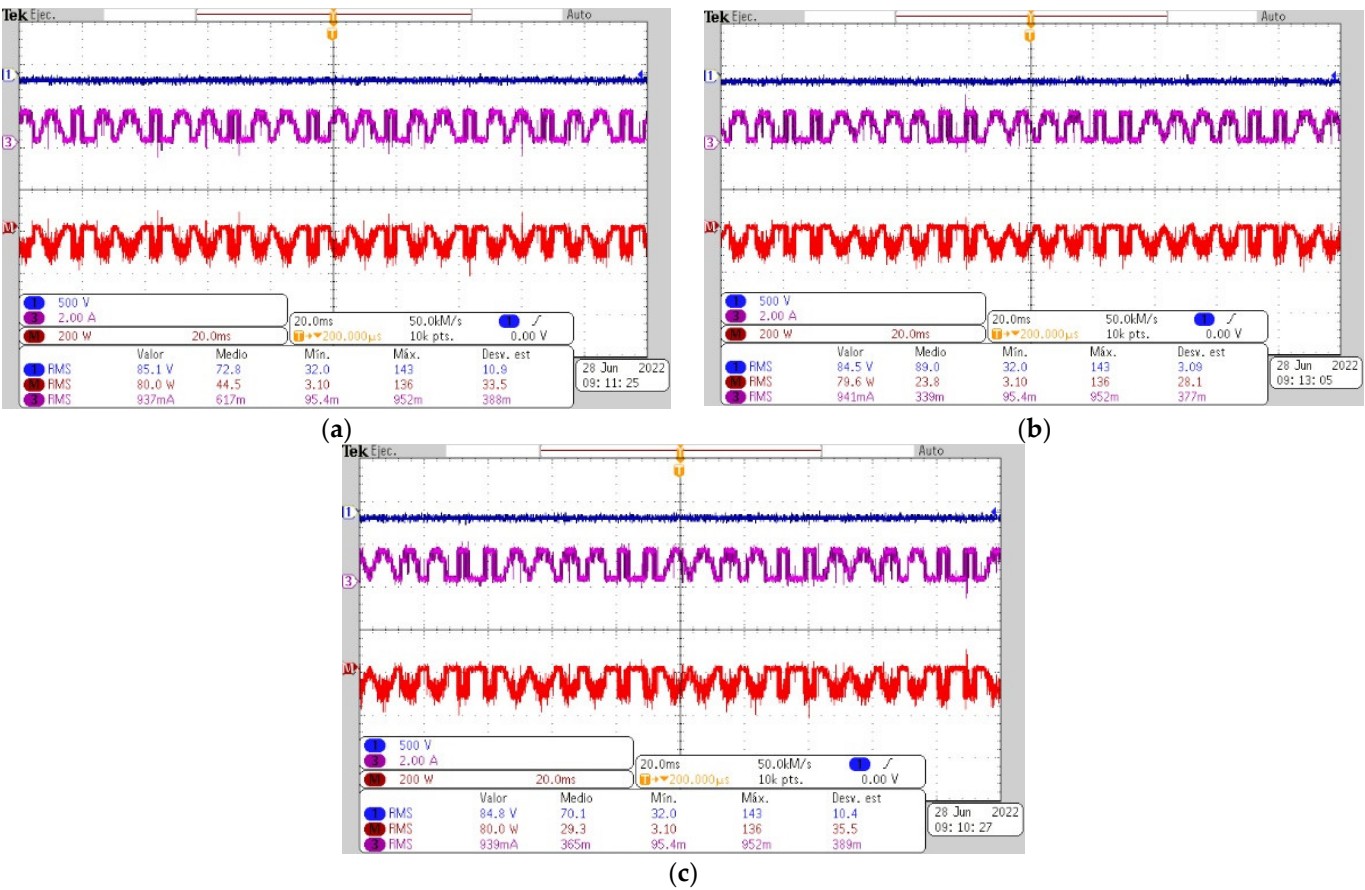

**Figure 17.** Signals of voltage (blue), power (red), and current (purple) in the cells: (**a**) cell 1; (**b**) cell 2; and (**c**) cell 2, employing the proposed alternative modulation strategy, "reconstructed modulators".

Finally, Figure 18 shows an amplification of the three voltage signals using the proposed alternative modulation strategy, which allows a better appreciation of its effect on the reconstructed modulating signals.

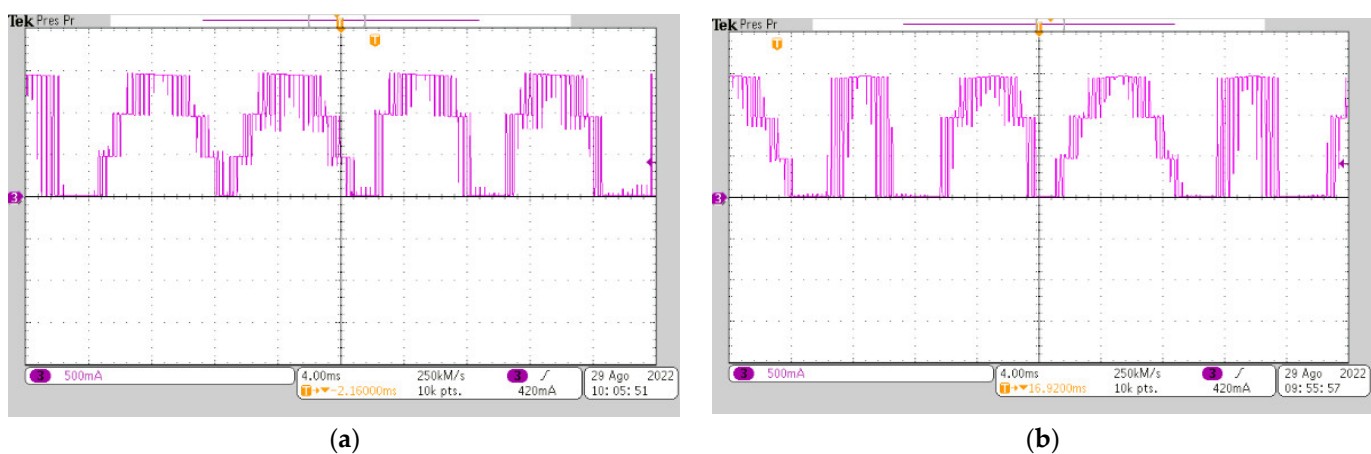

**Figure 18.** *Cont.*

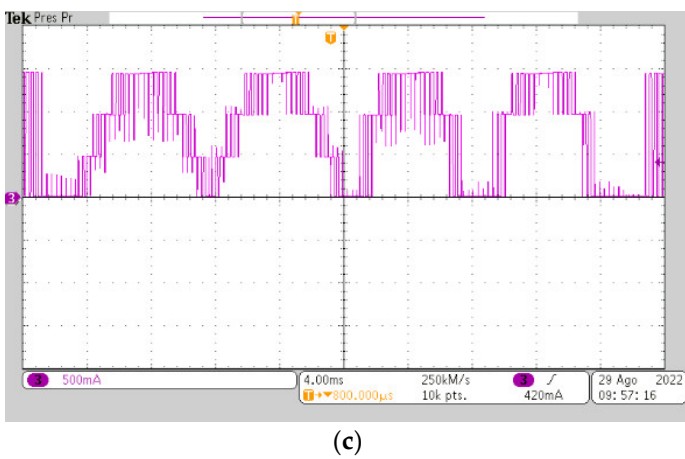

(**c**)

**Figure 18.** Amplification of the signals of voltage in the cells: (**a**) cell 1; (**b**) cell 2; and (**c**) cell 2, employing the proposed alternative modulation strategy, "reconstructed modulators".

## 4. Discussion

The following is a summary of the experimentally obtained results of the power-modulation set, considering the comparison parameters established in the previous sections.

Table 2 shows the behavior of the modulation strategies in terms of the use of logic elements and the percentage of digital resources. As can be seen in this table, the proposed modulation strategy uses fewer digital resources of the FPGA due to its construction and its use of fewer high-frequency carrier signals and more low-frequency ones. The energy balance strategy that uses more digital resources is the PSC. This is because this type of strategy multiplies the switching frequency according to the number of carriers and, consequently, there are more switches at the output.

**Table 2.** Topology design specifications.

| Parameter | Modulation Strategy Implemented | | | | |
|---|---|---|---|---|---|
| | PD | PSC | LS per carrier signal cycle | LS per modulating signal cycle | Alternative "reconstructed modulators" |
| Number of VHDL codes required | 11 | 11 | 14 | 14 | 9 |
| Combinational logic elements | 1231 | 2829 | 1197 | 1504 | 1667 |
| Dedicated logic registers | 94 | 132 | 93 | 105 | 73 |
| Pins | 7 | 7 | 7 | 7 | 7 |
| Percentage of digital resources used | 27% | 61% | 26% | 33% | 12% |

Table 3 shows the results obtained for the THD and distortion factor (DF) in implementation. It was observed that the alternative strategy proposed in this article obtained the lowest levels, with a 1.34% THD and 1.05% DF.

**Table 3.** THD and DF at the output voltage (m = 0.9).

| Modulation Strategy | THD (%) | DF (%) |
|---|---|---|
| PD | 3.15 | 2.64 |
| PSC | 2.38 | 1.99 |
| LS carrier | 3.35 | 2.81 |
| LS modulator | 2.67 | 2.08 |
| Alternative "reconstructed modulators" | 1.34 | 1.05 |

Regarding the percentage of power unbalance between cells, Table 4 shows the percentage of unbalance between cells under the different modulation strategies. Subsequently, Figure 19 shows the maximum power unbalance obtained in each cell. It is highlighted that

the proposed modulation strategy, "reconstructed modulators", is the one that presents the lowest inter-cell unbalance, and at the same time, it is verified that the PWM PD does not carry out power balancing.

**Table 4.** Percentage power unbalance between the cells of the different modulation strategies.

| Modulation Strategy | Percentage of Power Unbalance between Cells | | |
|---|---|---|---|
| | Cell 1–Cell 2 | Cell 1–Cell 3 | Cell 2–Cell 3 |
| PD | 49.62% | 59.04% | 18.68% |
| PSC | 16.33% | 1.09% | 16.89% |
| LS carrier | 9.23% | 9.23% | 0% |
| LS modulator | 0.66% | 6.62% | 6% |
| Alternative "reconstructed modulators" | 0.64% | 0.64% | 0% |

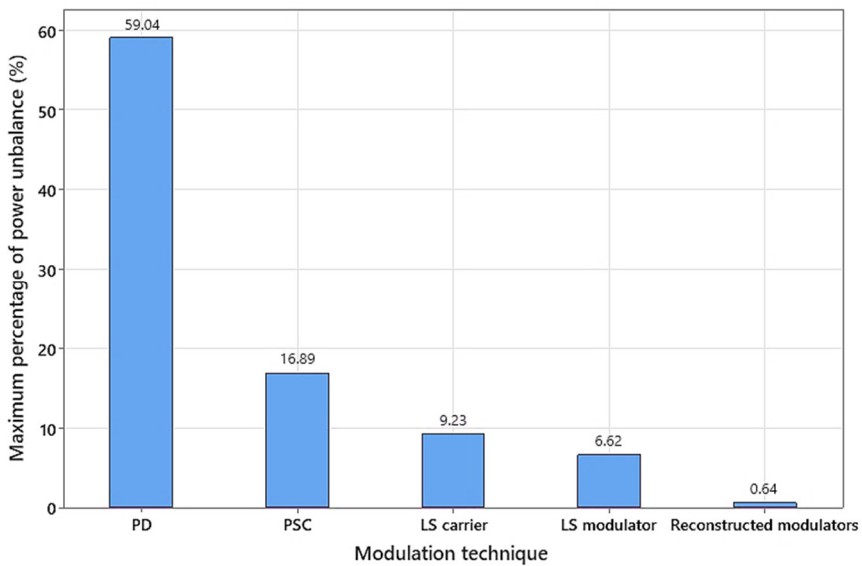

**Figure 19.** Maximum percentage of power unbalance between the cells.

Finally, Figure 20 shows the average energy obtained in each cell for the different strategies. In this graph, it can be confirmed that, as in the case of the simulation with the alternative strategy, more energy was transferred to the load, which was 20.05% more than the nearest value of the remaining strategies.

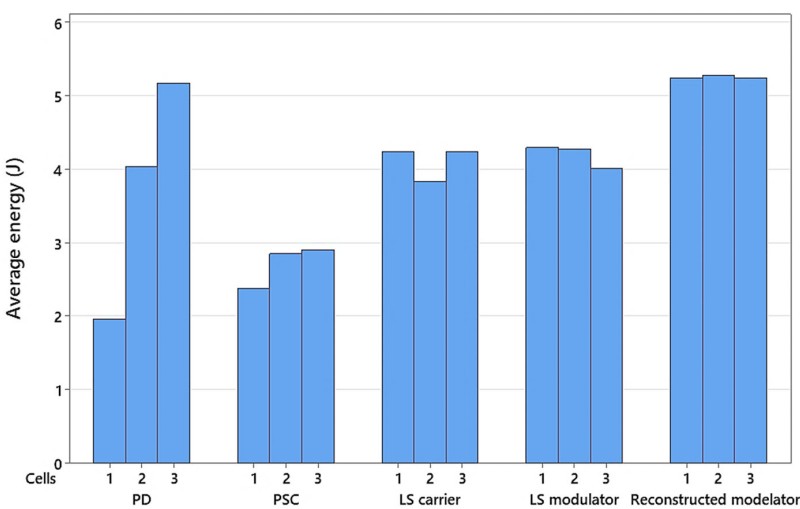

**Figure 20.** Average energy transferred per cell.

## 5. Conclusions

There are different parameters that evaluate the performance of inverters used in photovoltaic applications. One of them is the energy balance between the cells that make up the inverter. One of the alternatives to ensure that the cells of the cascaded multilevel converter are balanced is to modify the modulation strategy used in its switches.

For this reason, in this article, an alternative strategy was developed whose objective function was to carry out the energy balance between the cells of the cascade multilevel converter, and at the same time, to present better results in the comparison parameters of the existing strategies that comply with the energy balance.

The implemented alternative modulation strategy obtained the following results with respect to the strategies reported in the literature:

- a lower THD, obtaining 1.34%;
- a lower DF, obtaining 1.05%;
- a lower percentage of unbalance between the cells, obtaining 0.64%;
- a lower percentage of digital resource use, obtaining 12%; and
- a higher power transfer to the load of 20.05% more than the closest strategy.

**Author Contributions:** Conceptualization, S.E.D.L.A.; data curation, Y.R.-S.; formal analysis, Y.R.-S. and J.A.A.; funding acquisition, R.E.L.-P. and J.A.M.H.; investigation, Y.R.-S.; methodology, J.A.A.; project administration, S.E.D.L.A.; resources, L.M.C.-S. and R.E.L.-P.; software, L.M.C.-S. and J.A.M.H.; supervision, S.E.D.L.A. and J.A.A.; validation, J.A.A.; visualization, L.M.C.-S. and J.A.M.H.; writing—original draft, Y.R.-S. and S.E.D.L.A.; writing—review and editing, L.M.C.-S. and R.E.L.-P. All authors have read and agreed to the published version of the manuscript.

**Funding:** This research received no external funding.

**Institutional Review Board Statement:** Not applicable.

**Informed Consent Statement:** Not applicable.

**Data Availability Statement:** The data obtained in this study are available on request from the authors.

**Conflicts of Interest:** The authors declare no conflict of interest.

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
