# Peer review of "Modification of SPWM Modulating Signals for Energy Balancing Purposes"

_electronics, doi:10.3390/electronics11182871_

Round 1

Reviewer 1 Report

The paper presents alternative strategy designed reconstructed modulators for energy balance between cells of multilevel converters. The following comments are for the authors of the paper.

1.       In Figure 1, is it CD to AC?

2.       Figure 3 is not clear and some other figures in the paper.

3.       The theory and mathematical formulations in the paper are weak.

4.       The results are not rigorous enough to bring out the contribution of the paper. There are no comparisons with existing solutions in the literature.

5.       The conclusion is not well written.

Author Response

Editor in Chief

Electronics                                                                                                                     Aug29th, 2022

Dear Editor:

Subject: Submission of revised paper entitled "Modification of SPWM modulating signals for energy balancing purposes". Submission no: 1887399.

Thank you for your email dated Aug 19th, 2022 enclosing the reviewers’ comments. We have carefully reviewed the comments and have revised the manuscript accordingly. Our responses are given in a point-by-point manner below. Changes to the manuscript are highlighted in yellow color.

We hope the revised version is now suitable for publication and look forward to hearing from you in due course.

Sincerely,

Dr. Susana Estefany De León Aldaco

  • Reviewer 1
  • In Figure 1, is it CD to AC?

Response: Considering your comment. Figure 1 was corrected as follows:

  1. The abbreviation "CD" was corrected by "DC."
  2. A block called "filter" was added, at the suggestion of reviewer 3

  • Figure 3 is not clear and some other figures in the paper.

Response: Taking into account your comment. In order to improve the quality of figure 3, the simulation schematic diagram was recreated. In addition, the simulation was rerun to obtain figures 4 to 12, to achieve better quality and visibility.

  • The theory and mathematical formulations in the paper are weak

 Response: In response to this commentary, section 2.2.3 was modified, so that the mathematical formulations pertaining to the carrier signals and the reconstructed modulator signals proposed in this work were added. They are as follows:

This modulation strategy requires the use of only two carrier signals and three modulating signals to obtain a single-phase output voltage signal of seven levels. The signals used in this strategy are the following:

  • Carrier signal 1. Fixed-amplitude triangular signal located in the positive quadrant, see (2).
  • Carrier signal 2. Fixed amplitude triangular signal located in the negative quadrant, see (3).

(2)

(3)

Where, VC1 and VC2, are the voltage signals of the triangular carriers. ? is the period. fC is the frequency of the carrier signal. AT is the amplitude of the triangular waveform.

  • Modulating signal 1. The Sine signal is constructed in sections starting at an angle of 0°. Adding the angles (α), which set the level limits corresponding to each reconstructed modulator, see (4). The modulating signal 1 is the one that, when compared with the two carrier signals, obtains the switching signals of cell 1. The amplitude varies depending on the modulation index.

(4)

where, Vmod1, Vmod2, y Vmod3 are the voltage of each cell 1, 2, and 3, respectively. fm is the modulating frequency of the sine waveform. Vm is the amplitude of the sine waveform. Vpart is the voltage for each section of the modulator used.

  • Modulating signal 2. Sine signal is constructed in sections starting at an angle of 120°. Adding the angles (α), which set the level limits corresponding to each reconstructed modulator, see (5). The modulating signal 2 is the one that when compared with the two carrier signals obtains the switching signals of cell 2. The amplitude varies depending on the modulation index.

(5)

  • Modulating signal 3. Sine signal is constructed in sections starting at an angle of 240°. Adding the angles (α), which set the level limits corresponding to each reconstructed modulator, see (6). The modulating signal 3 is the one that when compared with the two carrier signals obtains the switching signals of cell 2. The amplitude varies depending on the modulation index.

(6)

  • The results are not rigorous enough to bring out the contribution of the paper. There are no comparisons with existing solutions in the literature.

Response: Dear reviewer, we understand your comment. The definition of the existing modulation strategies for energy balance purposes are described in section 2.2.2. Also, section 4 presents the comparison of the results of the proposed alternative strategy with respect to the existing strategies mentioned above.

In lines 157 and 162 the following paragraph has been added detailing the contribution of this article for greater clarity in the reading:

The previous section dealt with modulation strategies without energy balance purposes. We proceed to address the techniques whose objective function is to carry out the energy balance between the cells that make up the cascaded multilevel inverter. This to identify areas of opportunity for improvement under which the proposed alternative strategy will be governed. The above to finally carry out the comparison between these existing strategies and the alternative strategy proposed in this paper.

  • The conclusion is not well written.

Response: In response to this observation, the conclusions were rewritten. The conclusions were rewritten as follows:

There are different parameters that evaluate the performance of inverters used in photovoltaic applications. One of them is the energy balance between the cells that make up the inverter. One of the alternatives to ensure that the cells of the cascaded multilevel converter are balanced is to modify the modulation strategy used in its switches.

For this reason, in this article an alternative strategy was developed whose objective function is to carry out the energy balance between the cells of the cascade multilevel converter and, at the same time, to present better results in the comparison parameters of the existing strategies that comply with the energy balance.

 The implemented alternative modulation strategy obtained the following results with respect to the strategies reported in the literature:

- Lower THD, obtaining 1.34%;

- Lower DF, obtaining 1.05%;

- The lowest percentage of unbalance between cells, obtaining 0.64%;

- The lowest percentage of use of digital resources, obtaining 12%;

- Higher power transfer to the load, 20.05% more than the closest strategy.

Reviewer 2 Report

I have read the manuscript “Modification of SPWM modulating signals for energy balancing purposes”. The authors propose the alternative modulation strategy “reconstructed modulators”. They show that this alternative modulation strategy can realize energy balance in each cell of the multilevel inverter. They compare this alternative modulation strategy with the other four modulation strategies in detail, and find that this alternative modulation strategy can present better results than those of the other four modulation strategies.
The manuscript is well written. The research design is appropriate. The experimental methods are adequately described. The results are clearly laid out and are useful for the Electronics community. Thus, I recommend for
publication in Electronics.

Author Response

  • Reviewer 2:

I have read the manuscript “Modification of SPWM modulating signals for energy balancing purposes”. The authors propose the alternative modulation strategy “reconstructed modulators”. They show that this alternative modulation strategy can realize energy balance in each cell of the multilevel inverter. They compare this alternative modulation strategy with the other four modulation strategies in detail, and find that this alternative modulation strategy can present better results than those of the other four modulation strategies. The manuscript is well written. The research design is appropriate. The experimental methods are adequately described. The results are clearly laid out and are useful for the Electronics community. Thus, I recommend for publication in Electronics.

Response: Dear reviewer, we appreciate your valuable comments.

Reviewer 3 Report

very good work. Authors followed my suggestions in order to emprove the paper.

now it can be accepted without any problem

Round 2

Reviewer 1 Report

No comments. 

Reviewer 3 Report

I think that the paper named "Modification of SPWM modulating signals for energy balancing purposes" is interesting for research community, and I will like its publication.

The paper is well written and i followed it in a easili way.

It deals with an important issue and there is a lot of things to describe. I think that there are few pages, Authors can do better.

Suggestions:

1) figure 1, before the load there is a filter

2) line 46, should be cascaded h-bridge multilevel inverters (CHBMI)?

3) figure 2, I would put a capacitor between DC source and H-bridge

4) line 112, use DC

5) figure 3, change the quality, use a better legend, there is no description of resistive load.

6) on table 1, use the symbol of ohm, omega

7) I suggest to say that authors used only phase dispostion PD strategy, but there are others like in this article I suggest to consider "Experimental evaluation of the performance of a three-phase five-level cascaded h-bridge inverter by means FPGA-based control board for grid connected applications",Phase Disposition (PD); (b) Phase Opposition Disposition (POD); (c) Alternative Phase Opposition Disposition (APOD); (d) Phase Shifted (PS).

8) on lines 143 and 144, same acronyms are used, I know that is right, but authors can use the way that they used in table 4, LS carried and LS modulator

9) line 150 two dots are red

10) figure 5, there are no modulating signal

11) say that in figure 9 the modulating signal for each h-bridge, but in figure 10, mod 1 is red...

12) figure 10, Vmod1, should be bluem and fix the legend, there is a y not and

13) figure 12, say that the profile of voltage is withoud filters, a profile of current should be useful for a not resistive load

14) a better figure than 13 is required, a multilevel is greater than the one shown, there are the power element, the DC sources... check figure 31-32 of suggested article "Experimental evaluation of the performance of a three-phase five-level cascaded h-bridge inverter by means FPGA-based control board for grid connected applications"

15) figure 16 is very good, can I suggest to do as in figure 47 of suggested paper, a magnification is required at the same time, to let see the different signals named Vmod1, Vmod2 and Vmod3

16) the discussion is very useful. I found that there are some missin issues that authors can cite. The work is good for a resistive load, but no filter is designed, no current control is provided, dead time is not discussed, fault tolerant issue is not faced, how the system can overrun a h-bridge fault?

Good work!

Author Response

Editor in Chief

Electronics                                                                                                                     Aug29th, 2022

Dear Editor:

Subject: Submission of revised paper entitled "Modification of SPWM modulating signals for energy balancing purposes". Submission no: 1887399.

Thank you for your email dated Aug 19th, 2022 enclosing the reviewers’ comments. We have carefully reviewed the comments and have revised the manuscript accordingly. Our responses are given in a point-by-point manner below. Changes to the manuscript are highlighted in yellow color.

We hope the revised version is now suitable for publication and look forward to hearing from you in due course.

Sincerely,

Dr. Susana Estefany De León Aldaco

Reviewer 3:

I think that the paper named "Modification of SPWM modulating signals for energy balancing purposes" is interesting for research community, and I will like its publication. The paper is well written and i followed it in a easili way.It deals with an important issue and there is a lot of things to describe. I think that there are few pages, Authors can do better.

  • Figure 1, before the load there is a filter.

Response: Considering your comment. Figure 1 was corrected as follows:

  1. The acronymous "CD" was corrected by "DC”, at the suggestion of reviewer 1
  2. A block called "filter" was added.

  • Line 46, should be cascaded h-bridge multilevel inverters (CHBMI)?

Response:  In response to your comment. The sentence in lines 45 and 46 was rewritten as follows: “Among the multilevel topologies, the Cascaded H-bridge Multilevel Inverter topology (CHBMI) stands out,”

  • figure 2, I would put a capacitor between DC source and H-bridge

Response: Taking into account your comment. In figure 2 a capacitor block is added between the DC source and the multilevel inverter.

  • line 112, use DC

Response: Following your suggested comment. The abbreviation "dc" was changed to "DC", in capital letters.

  • figure 3, change the quality, use a better legend, there is no description of resistive load.

Response: Taking into account your comment. In order to improve the quality of figure 3, the simulation schematic diagram was recreated. In addition, the simulation was rerun to obtain figures 4 to 12, to achieve better quality and visibility.

The legend of figure 3 was also modified to read as follows: Schematic diagram of power stage: CHBMI and resistive load

  • on table 1, use the symbol of ohm, omega

Response: The omega symbol was modified in Table 1.

  • I suggest to say that authors used only phase dispostion PD strategy, but there are others like in this article I suggest to consider "Experimental evaluation of the performance of a three-phase five-level cascaded h-bridge inverter by means FPGA-based control board for grid connected applications",Phase Disposition (PD); (b) Phase Opposition Disposition (POD); (c) Alternative Phase Opposition Disposition (APOD); (d) Phase Shifted (PS).

Response:

It is important to emphasize that the PD strategy is a variant of the multicarrier PWM technique. Most variants of this are not for energy balancing purposes, such as: Phase Opposite Disposition (POD), Alternative Phase Opposite Disposition (APOD), which follow the same principle described above but vary by the phase of the carrier signals and each of them has a DC increment[14]. These variants are presented below:

  • In this modulation strategy, the carrier signals are in phase with each other.
  • In this modulation strategy the carrier signals are 180° out of phase with respect to the adjacent carrier signal.
  • In this modulation strategy the carrier signals above zero are 180° out of phase with respect to the carrier signals below zero.

Figure 4 shows the three variants of the carrier disposition PWM modulation strategy. Each modulation technique presented in this figure has six carrier signals, when compared with the modulating signals generate an output wave of seven levels. The carrier signals have the same amplitude, but different displacement.

(a)                                                                     (b)

(c)

Figure 4. Variants of PWM modulation strategy of carrier disposition: (a) PD (b) POD (c) PSC

  • on lines 143 and 144, same acronyms are used, I know that is right, but authors can use the way that they used in table 4, LS carried and LS modulator

Response: Derived from your suggestion concerning the acronyms of the level shifting strategy. The sentence in lines 144 and 145 was rewritten for better understanding, remaining as follows: LS PWM (Level Shifted PWM) with carrier level shift, in two variants: per modulator signal cycle (LS modulator) and per carrier signal cycle (LS carrier).

  • line 150 two dots are red

Response: Corrected the font color in the dots.

  • figure 5, there are no modulating signal

Response: Considering your comment. It was simulated again to obtain the modulating signal in conjunction with the carrier signals, corresponding to the PSC PWM strategy belonging to Figure 5. In addition, the following text was added before figure 5: "The frequency of the carrier signals present in figure 5 was reduced six times. This is only to better appreciate the phase difference between them".

  • say that in figure 9 the modulating signal for each h-bridge, but in figure 10, mod 1 is red...

Response: In response to your comment. The color correspondence in the modulating signals of the proposed strategy was corrected. The result is as follows:

*Modulating signal 1 (Vmod1) in red color.

*Modulating signal 2 (Vmod2) in blue color.

*Modulating signal 3 (Vmod3) in dark red color.

  • figure 10, Vmod1, should be bluem and fix the legend, there is a y not and

Response: Considering your previous comment. The color correspondence was established again, leaving the modulating signal 1 (Vmod1) in red. In addition, the typing error was corrected, changing "and" for "and" in the legend of figure 10.

  • figure 12, say that the profile of voltage is withoud filters, a profile of current should be useful for a not resistive load

Response: Dear reviewer, we agree with your comment. However, our case study was using a resistive load. It is for this reason that a filter was not used and designed.

Currently, we are taking your suggestions for future work in which we contemplate using other types of loads and contemplate the design and implementation of a filter.

  • a better figure than 13 is required, a multilevel is greater than the one shown, there are the power element, the DC sources... check figure 31-32 of suggested article "Experimental evaluation of the performance of a three-phase five-level cascaded h-bridge inverter by means FPGA-based control board for grid connected applications"

Response: In response to the suggestions and taking into account the example of Figures 31 and 32 of the article "Experimental evaluation of the performance of a three-phase five-level cascaded h-bridge inverter by means FPGA-based control board for grid connected applications", the description of the experimental platform was modified as follows:

This section deals with the results obtained experimentally. Figure 13 shows the ex-perimental platform developed. The power platform consists of two stages:

  • Stage 1. In order to ensure that the DC supply voltage is the same in the H-bridges, first the AC supply voltage of the network is transformed and then rectified to DC;
  • Stage 2. Once the power balance of the three DC sources is ensured, the single-phase cascaded multilevel inverter is used to perform the DC to AC conversion, integrated by IRAMS10UP60b modules.

Figures 14a and 14b shows a close-up of the the implementation of stage 1 and stage 2, respectively.     

Figure 13. Experimental platform, multilevel topology   

                          (a)                                                                             (b)

Figure 13. Power platform (a) stage 1 and (b) stage 2

  • figure 16 is very good, can I suggest to do as in figure 47 of suggested paper, a magnification is required at the same time, to let see the different signals named Vmod1, Vmod2 and Vmod3

Response: Based on your suggestion, the following paragraph and figure were added in lines 316 to 320:

Finally, Figure 17 shows an amplification of the three voltage signals using the proposed alternative modulation strategy. This is to better appreciate the effect of the reconstructed modulating signals.

(a)

(b)

(c)

Figure 17. Amplification of Signals of voltage in cells: (a)cell 1; (b) cell 2; (c) cell 2; employing the proposed alternative modulation strategy "reconstructed modulators”

  • the discussion is very useful. I found that there are some missing issues that authors can cite. The work is good for a resistive load, but no filter is designed, no current control is provided, dead time is not discussed, fault tolerant issue is not faced, how the system can overrun a h-bridge fault?

Response: In response to your comment. It is important to note that the work is limited to work in open loop, focusing on ensuring the energy balance between cells. It is for this reason that addressing the design of a current control and fault tolerance is beyond the scope of the work. However, your observation is opportune and that is why we will focus on addressing these issues in future work.
